# Clinical and Microbiologic Analysis of *Klebsiella pneumoniae* Infection: Hypermucoviscosity, Virulence Factor, Genotype, and Antimicrobial Susceptibility

**DOI:** 10.3390/diagnostics14080792

**Published:** 2024-04-10

**Authors:** Miri Hyun, Ji Yeon Lee, Hyun Ah Kim

**Affiliations:** Department of Infectious Diseases, Keimyung University Dongsan Hospital, Keimyung University School of Medicine and Institute for Medical Science, Keimyung University, Daegu 42601, Republic of Korea; eternity7919@dsmc.or.kr (M.H.); jirong84@dsmc.or.kr (J.Y.L.)

**Keywords:** *Klebsiella pneumoniae*, hypervirulence, hypermucoviscosity, aerobactin, community acquired liver abscess

## Abstract

Hypervirulent *Klebsiella pneumoniae* (KP) is defined according to hypermucoviscosity or various virulence factors and is clinically associated with community-acquired liver abscess (CLA). In this study, we investigated the clinical and microbiological characteristics of KP and significant factors associated with hypervirulence. The clinical characteristics, antimicrobial susceptibility, hypermucoviscosity, serotypes, hypervirulence-related genes, and biofilm formation of 414 KP isolates collected from the Keimyung University Dongsan Hospital between December 2013 and November 2015 were analyzed according to CLA. Significant risk factors for hypervirulent KP (HvKP) associated with CLA were investigated using logistic regression analysis. Notably, 155 (37.4%) isolates were hypermucoviscous, and 170 (41.1%) harbored aerobactin. CLA was present in 34 cases (8.2%). Epidemiology and treatment outcomes did not differ significantly between the CLA and non-CLA groups. The CLA group had significantly higher antibiotic susceptibility, K1/K2, *rmpA*, *magA*, *allS*, *kfu*, *iutA*, string test-positive result, and biofilm mass. Multivariate logistic regression revealed *rmpA* (OR, 5.67; 95% CI, 2.09–15.33; *p* = 0.001), *magA* (OR, 2.34; 95% CI, 1.01–5.40; *p* = 0.047), and biofilm mass >0.80 (OR, 2.13; 95% CI, 1.00–4.56; *p* = 0.050) as significant risk factors for CLA. *rmpA* was identified as the most significant risk factor for CLA among KP strains, implying that it is an important factor associated with HvKP.

## 1. Introduction

*Klebsiella pneumoniae* is a Gram-negative bacterium associated with an invasive syndrome that has caused liver abscesses in Southeast Asian populations over the past three decades [1,2,3]. Hypervirulent *K. pneumoniae* was first described in a patient with a liver abscess in Taiwan in the 1980s [4]. At that time, hypervirulence was defined based on hypermucoviscosity, which was confirmed by a string test, a phenotypic screening marker [5,6]. When hypervirulent *K. pneumoniae* first presented, it had the tendency to display low antimicrobial resistance, was associated with community-acquired infections, and was associated with the virulence factor *rmpA* and biofilm formation [4,7,8,9]. Hypervirulent *K. pneumoniae* has distinctive features compared with classic *K. pneumoniae*. Characteristics of classical *K. pneumoniae* infection were hospital-acquired infections, including pneumonia and urinary tract infections, and higher antimicrobial resistance rates [10,11].

Many studies on hypervirulent *K. pneumoniae* have been conducted over the years, and some researchers suggested that not only hypermucoviscosity but also aerobactin could be used to evaluate hypervirulence [12,13,14]. Aerobactin is a representative siderophore, and its production plays an important role in determining the hypervirulence of *K. pneumoniae* [15]. In several studies, the hypervirulent *K. pneumoniae* strains have been defined according to either aerobactin positivity or both hypermucoviscosity and aerobactin positivity [14,15,16].

In previous reports, the definition of hypervirulent *K. pneumoniae* varies depending on the definition used by each researcher, and there is still no consensus. Therefore, the proportion of hypervirulent *K. pneumoniae* inevitably depends on the definition used by each investigator [5,6,12,13,15,16,17]. For example, the prevalence of hypervirulent *K. pneumoniae* has been reported to vary from 12% to 58% in Southeast Asia [5,18,19,20,21,22].

Regarding the clinical aspect, hypervirulent *K. pneumoniae* is associated with community-acquired infections such as liver abscesses. Russo et al. [14] conducted a study in North America and the United Kingdom, defining hypervirulent *K. pneumoniae* based on the clinical diagnosis of tissue invasive infection. According to the results of the study, *peg-344*, siderophore-related virulence factors, and *rmpA* were some of the factors with the highest accuracy, sensitivity, and specificity in identifying hypervirulent *K. pneumoniae*. The string test had relatively low accuracy, sensitivity, and specificity (0.90, 0.89, and 0.91, respectively). K1 and K2 capsular serotypes had high specificity, but both accuracy and sensitivity were low [14]. The capsular serotypes of *K. pneumoniae*, such as K1 and K2, were reported to account for more than half of hypervirulent *K. pneumoniae* [9,23]. K1 and K2 serotypes have shown differences in clinical presentations, antimicrobial resistance rates, and molecular characteristics [18,24,25,26,27]. The K1 serotype mostly belongs to clonal group 23 (CG23), whereas the genetic characteristics of K2 are more diverse than those of the K1 serotype, and K2 belongs to diverse clonal groups, such as CG65, CG 86, and CG 375 [28,29,30]. Therefore, in this study, we attempted to focus on the relationship between community-acquired liver abscess (CLA) and hypervirulent *K. pneumoniae* from a clinical perspective in South Korea. The purpose of this study was to investigate the clinical and microbiological characteristics of *K. pneumoniae* and important factors related to its hypervirulence in South Korea.

## 2. Methods

### 2.1. Study Participants

A total of 414 *K. pneumoniae* isolates recovered from specimens collected at the Keimyung University Dongsan Hospital, a 1018-bed hospital in Daegu, South Korea, between November 2013 and November 2015 were retrospectively analyzed. We identified *K. pneumoniae* isolated from clinical specimens twice a week. When a *K. pneumoniae* strain was isolated, the researcher determined whether it was a true pathogen through chart review and collected the strain. Specimens obtained from patients younger than 18 years old were excluded. Patients who were transferred to other hospitals were excluded because we could not evaluate the treatment outcomes. Isolates were obtained from each patient during the first diagnosis of *K. pneumoniae* infection, and subsequent infections in the same patient were not included in this study. The time of the symptom onset was identified through chart review. The infections were further categorized into community-acquired, healthcare-associated, and nosocomial infections. Community-acquired infections were defined as those in which symptoms occurred within 48 h after visiting the hospital. However, patients with community-acquired infections and healthcare-associated risk factors were categorized as having healthcare-associated infections. Healthcare-associated risk factors included hospitalization within 90 days, hemodialysis, intravenous medication in outpatient clinics, or residency in long-term care facilities. Nosocomial infections were defined as those in which symptoms occurred 48 h after a patient was admitted. All isolates were subcultured in Luria–Bertani broth (Difco, Becton Dickinson, Sparks, MD, USA) and frozen at −70 °C until subsequent use.

### 2.2. Study Design

Medical records were retrospectively analyzed to identify underlying diseases, predisposing factors, antibiotics used within the last 3 months, previous hospitalization, antimicrobial susceptibility, clinical presentations, currently administered antibiotics, and treatment outcomes of the patients. Acute kidney injury was defined as an increase in serum creatinine level by >0.3 mg/dL within 48 h, an increase in serum creatinine level to >1.5 times baseline within 7 days, or a decrease in urine volume to <0.5 mL/kg/h over 6 h. The McCabe–Jackson score was used as the criterion to predict the survival of patients based on the prognosis of the underlying diseases, which were classified as rapidly fatal, ultimately fatal, and nonfatal [31]. The early treatment outcome was determined after 72 h of empirical antibiotic treatment. Early treatment outcomes were classified as complete response, partial response, or treatment failure. A complete response was defined as an improvement in both clinical conditions, and laboratory findings such as level of white blood cell or C-reactive protein. A partial response was defined as either an improved clinical condition or improved laboratory findings. Treatment failure was defined as the worsening of both clinical conditions and laboratory findings. Death of a patient due to *K. pneumoniae* infection or a complication of the infection within 30 days was defined as an infection-related 30-day mortality. We compared the strains based on their hypermucoviscous phenotype and hypervirulence-associated genes.

### 2.3. Definitions of Hypervirulent K. pneumoniae

In this study, we defined hypervirulent *K. pneumoniae* as CLA from a clinical perspective. We analyzed the clinical and microbiological characteristics according to the CLA to determine which genotypes and phenotypes were most associated with the CLA group. In addition, as reference data, the results of a comparative analysis between the two definitions based on hypermucoviscosity and aerobactin positivity, which were previously known as hypervirulent *K. pneumoniae* definitions, were presented in the supplementary tables. A previous study showed that the capsular serotype tended to have high specificity but low accuracy and sensitivity, and because the factors affecting *K. pneumoniae* hypervirulence may have a complex effect on each other, the capsular serotype was excluded during multivariate analysis [14].

### 2.4. Detection of K. pneumoniae Hypermucoviscosity

The string test was performed to determine the hypermucoviscous phenotype [32]. The string test was positive when a bacteriologic inoculation loop could generate a viscous string >5 mm in length by stretching bacterial colonies on an agar plate (Figure 1).

### 2.5. Polymerase Chain Reaction

Capsular serotypes and virulence factors, including *rmpA* (461), *magA* (1283), *allS* (764), *mrkD* (340), *ybtS* (242), *kfu* (638), and *iutA* (920), were identified using multiplex polymerase chain reaction (PCR). Strains were serotyped as K1, K2, K5, K20, K54, or K57 or as non-determined when a specific serotype could not be identified. The primers for *bla*_SHV-1a_ were used for the positive control reactions. Amplification was performed using a C1000 Thermal Cycler (Bio-Rad, Pleasanton, CA, USA). Crude DNA was prepared by the lysis of the colonies at 100 °C for 10 min in 500 mL of sterile distilled water, followed by centrifugation. The lysed supernatant was used for PCR. The PCR program comprised an initial activation step at 95 °C for 15 min, followed by 30 cycles of 94 °C for 30 s, 60 °C for 90 s, 72 °C for 60 s, and a final extension period at 72 °C for 10 min. The amplicons were separated via electrophoresis at 100 V for 2 h using a 2% agarose gel. Specific primers used to detect the alleles of the target gene sequences are listed in Appendix A.

### 2.6. Biofilm Formation

Biofilm mass was determined using a microtiter plate assay and based on optical density (OD, 570 nm). To measure biofilm formation, the bacterial suspensions were incubated in 96-well plates. After overnight incubation at 37 °C for 24 h, crystal violet was added, thereby staining the biofilm purple. Biofilm mass was then measured using the corresponding OD_570 nm_ of the supernatant following solubilization of crystal violet in 99% ethanol. We used Synergy/HTX Spectrophotometer (BioTek instrument, Inc. Winooski, VT, USA) for analysis of biofilm. Each strain was tested in triplicate, with a positive control of *K. pneumoniae* ATCC 700603 and a negative control of *K. pneumoniae* ATCC 13883.

### 2.7. Antimicrobial Susceptibility Test

The isolates were analyzed using an automated microbial identification (Vitek2 Gram-negative identification system; bioMerieux, Lyon, France) and susceptibility test system (Vitek2 AST-N224 system). Antimicrobial susceptibility profiles were determined based on the breakpoints recommended in the guidelines of the 2012 Clinical and Laboratory Standards Institute (CLSI). Extended-spectrum β-lactamase (ESBL) production was detected using an automated methodology based on the Vitek2 AST-N224 system, which was used to inoculate and incubate bacteria according to the manufacturer’s recommendations.

### 2.8. Statistical Analyses

Statistical analyses were performed using Statistical Package for the Social Sciences version 21.0 (IBM Corp., Armonk, NY, USA). Categorical variables were compared using the chi-squared test or Fisher’s exact test. For continuous variables, the normality of distribution was evaluated using the Kolmogorov–Smirnov test. The Mann–Whitney *U* test and independent *t*-test were performed for data that followed non-normal and normal distributions, respectively. Statistical significance was defined as *p* < 0.05. Risk factors for the CLA group were assessed and analyzed using logistic regression in the total group. An independent variable with *p* < 0.1 in the univariate analysis was included in the multivariate analysis, and a variable with a final *p* < 0.05 was considered a significant risk factor.

## 3. Results

### 3.1. Clinical and Microbiological Characteristics According to Hypermucoviscosity

In total, 414 *K. pneumoniae* isolates were included in this study (Figure 2). A hypermucoviscous phenotype was observed in 155 isolates (37.4%). Male sex was more strongly associated with the hypermucoviscous phenotype (69.0%) than with the string-negative group (54.4%) (*p* = 0.003). In underlying diseases, solid tumors were less associated with hypermucoviscous phenotype (*p* = 0.002). Based on the McCabe classification, ultimately, fatal disease was less frequently associated with the hypermucoviscous phenotype (*p* = 0.017). In predisposing factors, percutaneous catheter drainage (*p* = 0.006) and invasive procedures (*p* = 0.001) were associated with the hypermucoviscous phenotype.

In the infection category, the proportion of community-acquired infection was more associated with hypermucoviscous phenotype (*p* < 0.001). Nosocomial infection was more associated with the string-negative group (*p* < 0.001). Among infection sources, intra-abdominal infection was more strongly associated with the hypermucoviscous phenotype than with the string-negative group (*p* = 0.001), whereas urinary tract infection showed a weaker association (*p* < 0.001). Regarding clinical presentations, metastatic infection, severe sepsis and septic shock, concomitant bacteremia, and admission to the intensive care unit (ICU), no differences were observed between the two groups. There were also no significant differences in the treatment outcomes, treatment failure, infection-related 30-day mortality, or acute kidney injury (Appendix A).

The rates of resistance to ciprofloxacin (*p* < 0.001), cefazolin (*p* < 0.001), cefotaxime (*p* < 0.001), ceftazidime (*p* < 0.001), piperacillin/tazobactam (*p* < 0.001), aztreonam (*p* < 0.001), and trimethoprim/sulfamethoxazole (*p* < 0.001) were lower in the hypermucoviscous phenotype. ESBL-producing strains were lower in the hypermucoviscous phenotype (*p* < 0.001) (Appendix A).

Aerobactin positivity was identified in 122 hypermucoviscous phenotype isolates (78.7%) and 48 string-negative isolates (18.5%). Among the hypermucoviscous strains, 122 (78.7%) were *rmpA*-positive strains, and 48 (31.0%) were *magA*-positive. Biofilm formation did not differ significantly between the hypermucoviscous phenotype and string-negative groups (Appendix A).

### 3.2. Clinical and Microbiological Characteristics According to Aerobactin Positivity

Aerobactin was detected in 170 isolates (41.1%). In underlying diseases, solid tumor (*p* < 0.001), chronic renal disease (*p* = 0.040), and solid-organ transplant (*p* = 0.018) were less associated with hypermucoviscous phenotype. In predisposing factors, L-tube (*p* = 0.003), recent operation (*p* = 0.002), and prior ICU admission within 1 month (*p* = 0.026) were less associated with the hypermucoviscous phenotype.

The proportion of community-acquired infection was more associated with the aerobactin-positive group (*p* < 0.001). Nosocomial infection was more associated with the aerobactin-negative group (*p* < 0.001). Regarding the infection source, intra-abdominal infection was more strongly associated with the aerobactin-positive group (*p* = 0.023), whereas urinary tract infection showed a weaker association (*p* < 0.001). In terms of clinical presentations, metastatic infection, concomitant bacteremia, and admission to ICU, no differences were observed between the two groups. Similarly, no significant differences were observed in the treatment outcomes, treatment failure, infection-related 30-day mortality, or acute kidney injury (Appendix A).

The rates of resistance to ciprofloxacin (*p* < 0.001), cefazolin (*p* < 0.001), cefotaxime (*p* < 0.001), ceftazidime (*p* < 0.001), piperacillin/tazobactam (*p* < 0.001), aztreonam (*p* < 0.001), and trimethoprim/sulfamethoxazole (*p* < 0.001) were lower in the aerobactin-positive group. ESBL positivity was lower in the aerobactin-positive group (*p* < 0.001) (Appendix A).

The hypermucoviscous phenotype was detected in 122 (71.8%) aerobactin-positive and 33 (13.5%) aerobactin-negative isolates. Overall, *rmpA*-positive and *magA*-positive strains accounted for 159 (93.5%) and 60 (35.3%) isolates in the aerobactin-positive group, respectively. Biofilm formation did not differ significantly between the two groups (Appendix A). 

### 3.3. Epidemiology and Clinical characteristics According to Community-Acquired Liver Abscess

CLA was detected in 34 isolates (8.2%). In epidemiology, there was no significant difference in male sex and age between the CLA and non-CLA groups. In underlying diseases, solid tumors (*p* = 0.022) and neurological disease (*p* = 0.012) were more associated with the non-CLA group (*p* = 0.022). Based on the McCabe classification, ultimately, fatal disease was less frequently associated with the CLA group (*p* = 0.004). In predisposing factors, percutaneous catheter drainage (*p* < 0.001), L-tube (*p* = 0.015), and invasive procedures (*p* < 0.001) were associated with the CLA group. In others, such as recent operation (*p* = 0.019) and cases of prior ICU admission within 1 month (*p* = 0.014) were associated with the non-CLA group.

In the category of infection, all infections in the CLA group were community-acquired infections, as defined. Regarding the infection source, all infections in the CLA group were intra-abdominal infections, as defined. In terms of clinical presentations, metastatic infection, severe sepsis and septic shock, and admission to ICU, no differences were observed between the two groups. The rate of concomitant bacteremia was significantly higher in the CLA group than in the non-CLA group (61.8% vs. 31.8%; *p* < 0.001). There were no significant differences in treatment outcomes, treatment failure, infection-related 30-day mortality, or acute kidney injury (Table 1).

### 3.4. Antimicrobial Susceptibility and Microbiological Characteristics According to Community-Acquired Liver Abscess

Regarding antimicrobial susceptibility, most antibiotics such as amoxicillin/clavulanate, aztreonam, cefazolin, cefepime, cefotaxime, ceftazidime, ciprofloxacin, piperacillin/tazobactam, and trimethoprim/sulfamethoxazole had significantly lower resistance to *K. pneumonia* in the CLA group than in the non-CLA group. ESBL positivity was lower in the CLA group than in the non-CLA group (*p* < 0.001) (Table 2).

In the serotype, K1 (58.8% vs. 15.0%; *p* < 0.001) and K2 (26.5% vs. 13.2%; *p* = 0.042) were significantly higher in the CLA group than in the non-CLA group. In the virulence gene analysis, *rmpA* (82.4% vs. 36.1%; *p* < 0.001), *magA* (47.1% vs. 12.4%; *p* < 0.001), *allS* (52.9% vs. 16.4%; *p* < 0.001), *kfu* (58.8% vs. 31.2%; *p* = 0.001), and *aerobactin* (79.4% vs. 37.6%; *p* < 0.001) were significantly related to the CLA group compared with the non-CLA group.

Phenotype analysis showed that the CLA group had a higher string test-positive result, indicating greater hypermucoviscosity than the non-CLA group (73.5% vs. 34.2%; *p* < 0.001). When comparing the biofilm mass in the two groups, the rate of biofilm mass ≥0.80 was significantly higher in the CLA group than in the non-CLA group (52.9% vs. 30.3%; *p* = 0.007).

### 3.5. Significant Virulence Factors for Community-Acquired Liver Abscess in K. pneumoniae

As mentioned in the Methods Section, hypervirulent in *K. pneumoniae* was defined as CLA in this study. In the univariate logistic regression analysis, string test-positive (odds ratio [OR], 5.34; 95% confidence interval [CI], 2.4–11.78; *p* < 0.001), *iutA*-positive (OR, 6.39; 95% CI, 2.71–15.06; *p* < 0.001), *rmpA*-positive (OR, 8.28; 95% CI, 3.34–20.49; *p* < 0.001), *magA*-positive (OR, 6.30; 95% CI, 3.01–13.19; *p* < 0.001), *allS*-positive (OR, 5.73; 95% CI, 2.77–11.86; *p* < 0.001), and *kfu*-positive results (OR, 3.15; 95% CI, 1.54–6.45; *p* = 0.002) and biofilm mass >0.80 (OR, 2.58; 95% CI, 1.27–5.25; *p* = 0.009) were significant factors (Table 3).

The multivariate logistic regression analysis revealed that *rmpA* (OR, 5.83; 95% CI, 2.15–15.78, *p* = 0.001) was the most statistically significant risk factor for CLA, followed by *magA* (OR, 2.34; 95% CI, 1.01–5.40, *p* = 0.047) and biofilm mass >0.80 (OR, 2.13; 95% CI, 1.00–4.56, *p* = 0.050) (Table 3).

## 4. Discussion

In this study, we found that the proportion of hypermucoviscosity and aerobactin gene expression of *K. pneumoniae* varies widely. The proportion of hypervirulence differed depending on the used definitions of hypervirulent *K. pneumoniae* whether hypermucoviscosity and aerobactin positivity or not. Significant factors in CLA, for which the definition of hypervirulent *K. pneumoniae* was first proposed, were *rmpA*, *magA*, and biofilm mass.

When comparing the two groups according to the string test, the characteristics of hypervirulent *K. pneumoniae* were similar to those already known. As with other study results, hypermucoviscosity and aerobactin positivity were not consistent [33]. The hypermucoviscous capsule of hypervirulent *K. pneumoniae* is a key factor of hypervirulence [34]. Hypermucoviscous capsule contributes to reduced human cell binding and evasion of neutrophil-mediated phagocytosis. Because of this mechanism, the overproduction of hypermucoviscous capsules by hypervirulent *K. pneumoniae* has been reported as an important factor that aids bacterial dissemination and metastatic infections in the host [35]. Capsules could be affected by various environments and conditions [36]. Several studies have shown that the string test is not an appropriate method for assessing *K. pneumoniae* hypervirulence [13,15,23,37].

When comparing the two groups according to aerobactin results, the characteristics of hypervirulent *K. pneumoniae* were similar to already known [13]. Hypervirulent *K. pneumoniae* is associated with the possession of large virulence plasmids [37]. Iron acquisition, increased capsule production, K1/K2 capsular serotypes, and the colibactin toxin have been identified as the four microbiological and genotypic characteristics of hypervirulent *K. pneumoniae* [37]. Among various microbiologic factors, the ability to acquire iron is essential for bacterial growth [38]. High-affinity iron uptake systems contribute to the virulence of *K. pneumoniae* [39]. Aerobactin is located on a large virulence plasmid of *K. pneumoniae* that is not present in most classic *K. pneumoniae* strains [40]. Aerobactin mediates the virulence of *K. pneumoniae* and accounts for increased siderophore production under iron-limiting conditions by hypervirulent *K. pneumoniae* [38]. Therefore, some researchers have used aerobactin-positive strains to define hypervirulent *K. pneumoniae.* However, aerobactin-positive strains do not completely correlate with the existing hypermucoviscous phenotype [12,13,16,41].

Owing to changes in the medical environment, findings that deviate from the previously reported characteristics of hypervirulent *K. pneumoniae*, such as an increase in antimicrobial resistance and the relationship between healthcare-associated infections, have been reported [15,21,42]. In cases of our study when only aerobactin was confirmed positive, and the string test was negative, the antimicrobial resistance rate tended to be higher compared to aerobactin-positive cases; thus, it can be assumed that the characteristics of the hypermucoviscous phenotype may be lost upon exposure to antibiotics [43]. In a Chinese study, hypervirulence was defined as a positive result for aerobactin, and approximately 75% of hypervirulent and 18% of classic *K. pneumoniae* strains were found to be string test-positive [13]. In another Chinese study, hypervirulent *K. pneumoniae* was defined as having both a hypermucoviscous phenotype and a positive result for aerobactin. In that study, the prevalences of *rmpA* and *magA* were found to be 81.3% and 78.8% in hypervirulent *K. pneumoniae* and 17.9% and 61.1% in classic *K. pneumoniae*, respectively [16].

Hypervirulent *K. pneumoniae* was distinguished from classic *K. pneumoniae* based on the clinical presentation of a community-acquired pyogenic liver abscess with metastatic infections, such as endophthalmitis, central nervous system involvement, lung involvement, and antimicrobial susceptible pathogen. [3,44,45]. The capsular serotype, determined by surface antigens, has also been reported as an important factor that may affect the virulence of *K. pneumoniae* [46]. Capsular serotypes of K1 and K2 in liver abscess and biofilm formation were more frequently associated with hypervirulent *K. pneumoniae* [7,47,48,49]. In pyogenic liver abscesses of *K. pneumoniae*, aerobactin was more correlated than hypermucoviscosity [50]. K1 and K2 are the predominant capsular serotypes of hypervirulent *K. pneumoniae* [27,51], with K1 being the most common capsular serotype, followed by K2 [9,52]. In several studies, K1 and K2 showed different expression of virulence factors, especially *rmpA*, *magA*, and aerobactin, and displayed higher levels of biofilm formation than other capsular serotypes [25,26,53]. The K1 serotype is associated with siderophore iron acquisition systems and invasive infections [18], whereas K2 has a higher diversity of sequence types [26,54,55].

Hypermucoviscosity is still being used as a criterion for hypervirulent *K. pneumoniae* in many studies [5,21,32,38]. Among the virulence factors, *rmpA* is particularly known to affect capsule production in hypervirulent *K. pneumoniae* [56]. Hypervirulent *K. pneumoniae* with mutations in *rmpA* lose hypermucoviscous phenotype and show a strong reduction in virulence [57]. Proportions of hypermucoviscosity and *rmpA* in the aerobactin-positive cases were similar to those of another study, and the distribution of other virulence factors, such as *magA,* was confirmed to be diverse [58]. Hypermucoviscosity, *rmpA*, aerobactin, and serotype K1 are useful laboratory markers when suspecting community-acquired *K. pneumoniae* bacteremic liver abscess [59]. In cases of *K. pneumoniae* liver abscess in China, all strains were *rmpA*-positive, and two-thirds of strains were *magA*-positive [60]. *rmpA* is specifically correlated with abscess formation in hypermucoviscous *K. pneumoniae* strains [61].

This study had several limitations. First, it was conducted retrospectively in a tertiary hospital, which may introduce bias in the data interpretation. Second, this study was conducted at a single center. These factors can significantly influence the results and their generalizability. Third, the patients admitted to a tertiary hospital may exhibit more severe symptoms than those in a primary medical center. As patients who were transferred to other hospitals were excluded, there was a limitation in determining the overall condition of patients with *K. pneumoniae* infections who visited our hospital. Finally, the number of analyzed strains was relatively low.

Despite these limitations, our findings determined that both definitions were useful, as they showed clinical and microbiological features suggestive of hypervirulent *K*. *pneumoniae* and that the virulence gene *rmpA* was significantly higher in CLA with hypervirulent *K. pneumoniae*. Additional discussion is needed in the future regarding the definition of hypervirulent *K*. *pneumoniae*.

## Figures and Tables

**Figure 1 diagnostics-14-00792-f001:**
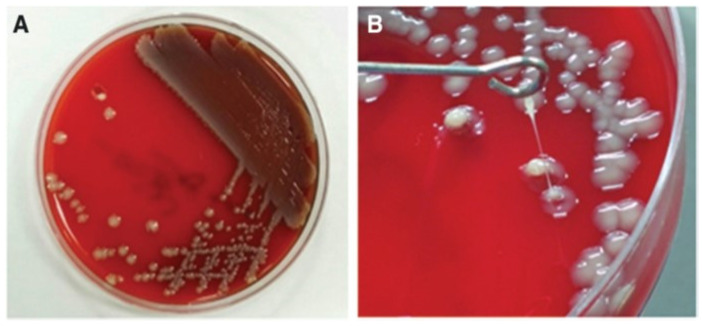
(**A**) Colonies are circular, convex with entire margin, and mucoid. (**B**) Stretching of the *K. pneumoniae* colonies resulted in the formation of a string ≥5 mm in length, demonstrating the hypermucoviscous phenotype.

**Figure 2 diagnostics-14-00792-f002:**
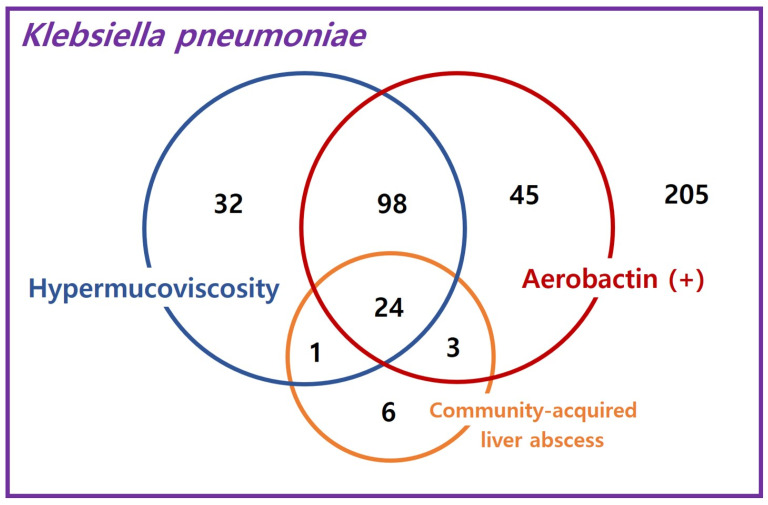
Distributions of the hypermucoviscosity, aerobactin positivity, and community-acquired liver abscess of *K. pneumoniae* in this study group.

**Table 1 diagnostics-14-00792-t001:** Baseline characteristics and clinical presentations of *Klebsiella pneumoniae* isolates according to CLA.

	CLA (−) (*n* = 380)	CLA (+) (*n* = 34)	*p* Value
Epidemiology			
Male sex	223 (58.7)	25 (73.5)	0.091
Age (years)	67.9 ± 13.6	66.6 ± 10.9	0.577
Underlying diseases			
Solid tumor	115 (30.3)	4 (11.8)	0.022
Chronic liver disease	45 (11.8)	5 (14.7)	0.585
Neurological disease	162 (42.6)	7 (20.6)	0.012
Chronic renal disease	39 (10.3)	0 (0.0)	0.060
Diabetes mellitus	133 (35.0)	13 (38.2)	0.705
Chronic lung disease	40 (10.5)	2 (5.9)	0.558
Solid-organ transplantation	12 (3.2)	0 (0.0)	0.610
McCabe classification			
Nonfatal underlying disease	282 (74.2)	33 (97.1)	0.004 *
Ultimately fatal disease	95 (25.0)	1 (2.9)	
Fatal disease	3 (0.8)	0 (0.0)	
Predisposing factors			
Urinary catheter	184 (48.4)	11 (32.4)	0.072
Percutaneous catheter drainage	62 (16.3)	26 (76.5)	<0.001
L-tube	92 (24.2)	2 (5.9)	0.015
Invasive procedure	91 (23.9)	26 (76.5)	<0.001
Recent operation	72 (18.9)	1 (2.9)	0.019
Prior ICU admission within 1 month	55 (14.5)	0 (0.0)	0.014 *
Category of infection			
Community-acquired infection	133 (35.0)	34 (100.0)	<0.001
Healthcare-associated infection	80 (21.1)	0 (0.0)	
Nosocomial infection	167 (43.9)	0 (0.0)	
Infection source			
Urinary tract infection	88 (23.2)	0 (0.0)	0.002
Intra-abdominal infection	38 (10.0)	34 (100.0)	<0.001
Respiratory infection	163 (42.9)	0 (0.0)	<0.001
Clinical presentation			
Severe sepsis and septic shock	125 (32.9)	9 (26.5)	0.443
Metastatic infection	4 (1.1)	1 (2.9)	0.351
Concomitant bacteremia	121 (31.8)	21 (61.8)	<0.001
Mechanical ventilation	64 (16.8)	3 (8.8)	0.224
Admission to ICU	100 (26.3)	6 (17.6)	0.267
Treatment outcomes			
Treatment failure (72 h)	68 (17.9)	2 (5.9)	0.073
Infection-related 30-day mortality	42 (13.7)	2 (10.0)	1.000
Acute kidney injury	46 (12.1)	5 (14.7)	0.592

Values are presented as mean ± standard deviation or number (%). * Fisher’s exact test. CLA: community-acquired liver abscess; ICU: intensive care unit.

**Table 2 diagnostics-14-00792-t002:** Antimicrobial resistance and microbiological characteristics of *Klebsiella pneumoniae* isolates according to CLA.

	CLA (−) (*n* = 380)	CLA (+) (*n* = 34)	*p* Value
Antimicrobial resistance rates			
Amikacin	19 (5.0)	0 (0.0)	0.387
Amoxicillin/clavulanate	122 (32.1)	1 (2.9)	<0.001
Aztreonam	136 (35.8)	1 (2.9)	<0.001
Cefazolin	139 (36.6)	1 (2.9)	<0.001
Cefepime	134 (35.3)	1 (2.9)	<0.001
Cefotaxime	136 (35.8)	1 (2.9)	<0.001
Ceftazidime	136 (35.8)	1 (2.9)	<0.001
Ciprofloxacin	111 (29.2)	0 (0.0)	<0.001
Ertapenem	0	0	n/a
Gentamicin	70 (18.4)	2 (5.9)	0.065
Imipenem	0	0	n/a
Piperacillin/tazobactam	98 (25.8)	2 (5.9)	0.009
Tigecycline	43 (11.3)	1 (2.9)	0.156
Trimethoprim/sulfamethoxazole	101 (26.6)	0 (0.0)	<0.001
ESBL positivity	134 (35.3)	1 (2.9)	<0.001
Serotype			
K1	57 (15.0)	20 (58.8)	<0.001
K2	50 (13.2)	9 (26.5)	0.042 *
K5	3 (0.8)	0 (0.0)	1.000
K20	19 (5.0)	0 (0.0)	0.387
K54	4 (1.1)	0 (0.0)	1.000
K57	16 (4.2)	0 (0.0)	0.383
ND	231 (60.8)	5 (14.7)	<0.001
Virulence gene			
*rmpA*	137 (36.1)	28 (82.4)	<0.001
*magA*	47 (12.4)	16 (47.1)	<0.001
*allS*	62 (16.4)	18 (52.9)	<0.001
*mrkD*	370 (97.9)	34 (100.0)	1.000
*entB*	374 (98.9)	34 (100.0)	1.000
*kfu*	118 (31.2)	20 (58.8)	0.001
*Aerobactin*	143 (37.6)	27 (79.4)	<0.001
String test	130 (34.2)	25 (73.5)	<0.001
Biofilm mass	0.67 ± 0.46	0.98 ± 0.77	0.026
>0.58 (median)	177 (48.4)	21 (61.8)	0.135
>0.80	111 (30.3)	18 (52.9)	0.007

Values are presented as *n* (%) or mean ± standard deviation. * Fisher’s exact test. CLA: community-acquired liver abscess; n/a: not available; ESBL: extended-spectrum β-lactamase; ND: not detected.

**Table 3 diagnostics-14-00792-t003:** Significant virulence factors for community-acquired liver abscess in *Klebsiella pneumoniae* using logistic regression analysis.

Variable	Univariate Analysis			Multivariate Analysis		
	OR	95% CI	*p* Value	OR	95% CI	*p* Value
String test (+)	5.34	2.42–11.78	<0.001	−	−	−
*iutA* (+)	6.39	2.71–15.06	<0.001	−	−	−
*rmpA* (+)	8.28	3.34–20.49	<0.001	5.83	2.15–15.78	0.001
*magA* (+)	6.30	3.01–13.19	<0.001	2.34	1.01–5.40	0.047
*allS* (+)	5.73	2.77–11.86	<0.001	−	−	−
*kfu* (+)	3.15	1.54–6.45	0.002	−	−	−
Biofilm > 0.80	2.58	1.27–5.25	0.009	2.13	1.00–4.56	0.050

Multivariate logistic regression analysis was performed using the backward-conditional method for factors with *p* < 0.1 in the univariate logistic regression analysis. OR: odds ratio; CI: confidence interval.

## Data Availability

The dataset of the current study is available from the corresponding author upon request.

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
