# Peer review of "Clinical and Microbiologic Analysis of Klebsiella pneumoniae Infection: Hypermucoviscosity, Virulence Factor, Genotype, and Antimicrobial Susceptibility"

_diagnostics, 2024, doi:10.3390/diagnostics14080792_

Round 1

Reviewer 1 Report

Comments and Suggestions for Authors

The manuscript of Hyun et al. “Clinical and ...” is interesting, but needs some corrections.

1.       In line 10 KP for Klebsiella pneumoniae was added, so why is it not used in the next lines?

2.       Line 30 “gram” should be written in capital - Gram

3.       Line 51 after Russo et al. citation should be added

4.       Study participants. The past tense should be used

5.       Line 89 Luria –Bertani company and country should be added

6.       Line 130 Why is aerobactin in italic? When you write the names of the genes like rmpA, magA,  aerobactin there should be an aerobactin gene too, not a word meaning virulence factor

7.       Line 143 Which analyzer the Authors used?

8.       Line 153 CLSI edition, year should be added

9.       Line 153 extended-spectrum β-lactamase should be changed to extended-spectrum β-lactamases

10.   Line 153  acronym ESβLs

11.   Greek letter for β-lactamase should be changed in whole manuscript

12.   Line 193 ESβL positivity? it should be changed

13.   Table 2 What for did the Authors estimate susceptibility to cefoxitin? This antibiotic is not used in the treatment of Enterobacterales infections

14.   Table 2 There is no information about susceptibility to meropenem, why? In many cases in “ESβL-positive infections” there is a drug of choice for the treatment

15.   Line 268 K. pneumoniae starts with capital letter

16.   There is no information about multidrug resistant strains only ESβL, were all ESβL-positive strains multidrug resistant?

17.   Line 288 “the Korean population”? Strains were isolated only from one hospital. It should be rearranged.

18.   Lines 312/326/332 etc. “classic” what does it mean?

19.   Lines 315/316 the phrase “aerobactin-positivity” should be changed

Reviewer 2 Report

Comments and Suggestions for Authors

The manuscript on phenotypic and genotypic characterization of hypervirulent K. pneumoniae focuses on the virulent aspects and antibiotic profiles. Useful markers like rmpA, aerobactin, K1 serotypes can be  considered  while suspecting community  acquired K.pneumoniae bacterimic liver abscess .This study can be used as a pilot study for future research work and would be a great surveillance resource ,
